# Conservative Axillary Surgery May Prevent Arm Lymphedema without Increasing Axillary Recurrence in the Surgical Management of Breast Cancer

**DOI:** 10.3390/cancers15225353

**Published:** 2023-11-09

**Authors:** Masakuni Noguchi, Masafumi Inokuchi, Miki Yokoi-Noguchi, Emi Morioka, Yusuke Haba

**Affiliations:** Department of Breast and Endocrine Surgery, Breast Center, Kanazawa Medical University Hospital, Daigaku-1-1, Uchinada, Kahoku 920-0293, Ishikawa, Japan; inokuchi@kanazawa-med.ac.jp (M.I.); miki-y@kanazawa-med.ac.jp (M.Y.-N.); emi-mori@kanazawa-med.ac.jp (E.M.); haba@kanazawa-med.ac.jp (Y.H.)

**Keywords:** axillary dissection, axillary reverse mapping, breast cancer, conservative axillary surgery, conservative axillary regional excision, partial lower axillary dissection, sentinel lymph node biopsy, tailored axillary surgery

## Abstract

**Simple Summary:**

Axillary lymph node dissection (ALND) has traditionally been performed to assess nodal status, prevent axillary recurrence, and possibly improve survival. However, the procedure has been associated with postoperative morbidities, including arm lymphedema, shoulder dysfunction, and paresthesia. Sentinel lymph node (SLN) biopsy was introduced as an alternative approach to assess axillary nodal status and potentially eliminate the need for ALND in patients with clinically node-negative (cN0) breast cancer. Despite this progress, eliminating ALND for all breast cancer patients still seems premature at this time. Recently, various forms of conservative axillary surgery have been developed to replace or supplement conventional ALND. Conservative axillary surgery may be promising in reducing the incidence of arm lymphedema without increasing the risk of axillary recurrence.

**Abstract:**

Axillary lymph node dissection (ALND) has been associated with postoperative morbidities, including arm lymphedema, shoulder dysfunction, and paresthesia. Sentinel lymph node (SLN) biopsy emerged as a method to assess axillary nodal status and possibly obviate the need for ALND in patients with clinically node-negative (cN0) breast cancer. The majority of breast cancer patients are eligible for SLN biopsy only, so ALND can be avoided. However, there are subsets of patients in whom ALND cannot be eliminated. ALND is still needed in patients with three or more positive SLNs or those with gross extranodal or matted nodal disease. Moreover, ALND has conventionally been performed to establish local control in clinically node-positive (cN+) patients with a heavy axillary tumor burden. The sole method to avoid ALND is through neoadjuvant chemotherapy (NAC). Recently, various forms of conservative axillary surgery have been developed in order to minimize arm lymphedema without increasing axillary recurrence. In the era of effective multimodality therapy, conventional ALND may not be necessary in either cN0 or cN+ patients. Further studies with a longer follow-up period are needed to determine the safety of conservative axillary surgery.

## 1. Introduction

Axillary lymph node dissection (ALND) has traditionally been performed to assess nodal status, prevent axillary recurrence, and possibly improve survival. However, the procedure has also been associated with postoperative morbidities, including arm lymphedema, shoulder dysfunction, and paresthesia [1]. The risk of arm lymphedema, in particular, has been used as an argument against ALND. Sentinel lymph node (SLN) biopsy was introduced as an alternative approach to assess axillary nodal status and potentially eliminate the need for ALND in patients with clinically node-negative (cN0) breast cancer [2,3]. Several randomized clinical trials have shown that ALND can be avoided not only in patients with negative SLN(s) [4,5] but also in selected patients with one to two positive SLN(s) undergoing either breast-conserving surgery (BCS) with whole breast irradiation [6] or mastectomy with axillary radiation [7]. Moreover, ALND can potentially be avoided in clinically node-positive (cN+) patients who convert to SLN-negative status after undergoing neoadjuvant chemotherapy (NAC) [8]. Despite this progress, eliminating ALND for all breast cancer patients still seems premature at this time. There are subsets of patients in whom ALND remains necessary [9].

Recently, various forms of conservative axillary surgery have been developed to replace or supplement conventional ALND. Conservative axillary surgery appears promising in reducing the incidence of arm lymphedema without increasing the risk of axillary recurrence. This article reviews the current indications for ALND in operable breast cancer and discusses the feasibility of conservative axillary surgery to decrease the incidence of arm lymphedema without increasing axillary recurrence.

## 2. Conventional ALND for cN0 and cN+ Patients

In 1907, Halsted demonstrated that radical mastectomy (including ALND) improved 3-year survival rates to ≧40% in comparison with no surgery [10]. Thereafter, ALND became the standard treatment for patients with operable breast cancer. However, in the 1970s, the National Surgical Adjuvant Breast and Bowel Project (NSABP) conducted their B-04 trial to investigate whether ALND could be avoided in patients with operable breast cancer (Table 1). The trial included a total of 1079 cN0 patients who were randomized to undergo radical mastectomy, total mastectomy with axillary radiation, or total mastectomy with delayed ALND if axillary metastases developed. After a follow-up period of 25 years, there was no significant difference in overall survival among the three groups [11]. Furthermore, while 44.6% of the patients who underwent ALND had nodal metastases, only 18.5% of the patients who did not receive ALND or axillary radiation developed axillary recurrence and subsequently required delayed ALND [11].

The B-04 trial also investigated treatment outcomes in cN+ patients. A total of 586 patients with cN+ breast cancer were randomized to undergo either radical mastectomy or total mastectomy with axillary radiation. As with the cN0 patients, there was no significant difference in overall survival between the two cN+ groups. However, rates of axillary recurrence were significantly higher in patients who received axillary radiation compared with those who underwent ALND (12% vs. 1%, *p* < 0.01) [12]. Although ALND did not contribute to a survival benefit in either cN0 or cN+ patients, it was effective to prevent axillary recurrence in both groups. Consequently, the NSABP B-04 trial did not lead to the abandonment of ALND for cN0 and cN+ patients. ALND continued to be performed as the standard method for assessing nodal status and preventing axillary recurrence. Nevertheless, results from the trial suggested that axillary radiation is more effective for achieving local control in cN0 patients than in cN+ patients [1].

## 3. Sentinel Lymph Node (SLN) Biopsy for cN0 Patients

ALND is unnecessary in patients without axillary metastases. SLN biopsy emerged as a method to assess axillary nodal status and possibly obviate the need for ALND in patients with cN0 breast cancer [2,3]. Randomized clinical trials have shown that SLN biopsy alone does not decrease survival or increase axillary recurrence when compared with ALND in SLN-negative patients (Table 2) [4,5]. Moreover, in a randomized clinical trial, there was a lower incidence of arm lymphedema in patients who underwent SLN biopsy alone compared with those who underwent ALND (6% vs. 37%) at 24 months post-surgery [13]. Thus, SLN biopsy can eliminate ALND in SLN-negative patients. Nevertheless, recently, axillary staging has not been indicated for staging, local control, or determining the need for systemic therapy in 70 or more older women with cT1-2, cN0, or luminal-type breast cancer [14].

While SLN biopsy has helped spare cN0 patients from unnecessary ALND, for SLN-positive patients, performing ALND has remained the standard method of preventing axillary recurrence. However, ALND is not always necessary in SLN-positive patients [15]. Randomized trials have shown that ALND can be safely avoided in selected cN0 patients with one or two positive SLNs undergoing either BCS with breast radiation [6] or mastectomy with axillary radiation (Table 3) [7]. In these patients, although low-volume residual disease may remain in the axilla after removing SLNs, it can be treated with postoperative radiotherapy and systemic therapy. The other randomized trials also supported these findings [16,17]. Nevertheless, ALND is still needed in patients with three or more positive SLNs or those with gross extranodal or matted nodal disease, as there is a higher risk of high-volume residual disease remaining in the axilla after the removal of SLNs [6,7].

## 4. Neoadjuvant Chemotherapy (NAC) for cN+ Patients

ALND has conventionally been performed to establish local control in cN+ patients with a heavy axillary tumor burden [9]. In fact, for cN+ patients, beyond ALND, options are still limited. The sole method to avoid ALND is through neoadjuvant chemotherapy (NAC). However, it also hinges on achieving a nodal pathologic complete response (pCR), as determined by the selective removal of axillary lymph nodes (ALNs) [8]. Indeed, it has been shown that ALND can be avoided in cN+ patients who become cN0 and SLN-negative after NAC [8]. However, SLN biopsy has a >10% false-negative rate (FNR) when performed after NAC [18,19]. In the SENTINA study [18], the detection rate of SLNs was 80.1% and the overall FNA was 14.2%; in the ACOSOG-Z1071 trial [19], the detection rate of SLNs was 92.9% and the FNA was 12.6%. On the other hand, targeted axillary dissection (TAD) may provide more accurate assessment of pathological response after NAC [20]. The technique for TAD is briefly described as follows [20]: an iodine-125 seed was placed in the clipped axillary node under ultrasound guidance before surgery. A radioisotope (technetium-99m sulfur colloid) and/or blue dye were injected into breast before or at the time of surgery. During surgery, a gamma probe on the iodine-125 setting was used to identify the seed-containing nodes and the technetium-99m setting was used to identify SLNs. All nodes containing blue dye, radioactivity, or which were palpable were removed as SLNs. This procedure was performed on patients with cN1 after NAC. TAD followed by ALND was performed with an FNR of 2.0% [20]. TAD is an axillary staging technique whereby the lymph node-positive for metastatic disease at initial diagnosis is marked using different methods such carbon tattooing, radioiodine, metallic clips, ferromagnetic seeds, etc., prior to NAC so that this marked lymph node can be removed during breast cancer surgery. If the marked lymph node cannot be identified or remains positive for metastatic disease after NAC, ALND is usually carried out [21,22].

Nevertheless, achieving pCR is highly dependent on tumor biology. NAC has proven to be extremely effective in patients with HER2-positive or triple-negative type breast cancers, resulting in nodal pCR in 44–78% of cases [23,24]. However, nodal pCR occurs only in about 20% of patients with luminal breast cancer. In this patient subgroup, which accounts for the majority of breast cancer cases, the use of NAC rarely eliminates the need for ALND [8,25]. In patients treated with NAC, residual nodal disease in the axilla potentially represents a population of tumor cells resistant to chemotherapy. The current standard of care for treating any residual tumor cells in the SLN after NAC is ALND [26].

## 5. Current Indications for ALND

The majority of breast cancer patients are eligible for SLN biopsy only, so ALND can be avoided. However, there is subsets of patients in whom ALND remains necessary [9]. According to the National Comprehensive Cancer Network (NCCN) Guidelines, ALND of level I and II nodes is recommended to patients with biopsy proven axillary metastases (in those who did not receive NAC) or who have residual disease after NAC [27]. However, from the ACOSOG Z0011 data [6], no ALND is recommended only if all of the following criteria are met: the patients have cT1-2, cN0 tumors; have not received NAC; only have one or two positive SLNs (macrometastases); and will undergo BCS with whole breast radiation. Moreover, in the St. Gallen International Consensus conference in 2021 [28], there was controversy about lower residual disease after NAC (for instance, a micrometastasis or isolated tumor cells in one of three SLNs). Axillary radiation could be an alternative to ALND in such a situation. On the other hand, Beck and Morrow [9] have summarized the current indications for ALND as follows. First, ALND is recommended as an axillary staging procedure for the following patients: those with cT3-4 tumors undergoing upfront surgery, those diagnosed with inflammatory breast cancer, those who remain cN+ (both in the upfront surgery setting or following NAC), and those with cN2-3 disease, regardless of their response to NAC. ALND is also indicated for establishing local control in patients with a high burden of axillary disease that has failed to be adequately managed by systemic therapy or radiotherapy. This category includes patients with three or more positive SLNs, those with gross extranodal or matted nodal disease as identified by intraoperative nodal palpation during upfront surgery, those with any residual nodal tumor following NAC, those with inflammatory breast cancer, and those with axillary recurrence [9].

## 6. Conservative Axillary Surgery

In the era of effective multimodality therapy, various forms of conservative axillary surgery have been developed to decrease the incidence of arm lymphedema without increasing axillary recurrence [29]. Conservative axillary surgery includes partial lower ALND, conservative axillary regional excision (CARE), conservative ALND with axillary reverse mapping (ARM), and tailored axillary surgery (TAS).

### 6.1. Partial Lower ALND for cN0 Patients

The axilla is anatomically divided into upper and lower parts, as separated by the second intercostobrachial nerve (ICBN) (Figure 1) [30,31]. ALNs and lymphatics draining from the upper extremity are generally located in the upper area situated between the axillary vein and the second ICBN [32], although lymph nodes lateral to the lateral thoracic vein are more specific to the lymphatic drainage of the upper extremity than to the drainage of the breast [33]. In a preliminary study of Kodama et al. [34], ALNs were excised separately from the upper and lower regions of the axilla in 100 cN0 breast cancer patients. On histological examination, nodal metastases were identified in the upper region in only 8% of patients. As a result, Kodama et al. subsequently developed a procedure known as partial lower ALND to preserve the ALNs and lymphatics located between the second ICBN and axillary vein. In this procedure, the level I axillary nodes caudal to the nerve were excised while retracting the second ICBN using tape. This is different from four-node sampling, where hard or enlarged axillary nodes were selectively removed from the axillary tissue. Such an approach was used in 1043 cN0 patients, with 248 (23.8%) patients undergoing total mastectomy and 795 (76.2%) patients undergoing BCS with breast radiation. A median of seven ALNs were removed per patient, and ALNs were involved in 21.3% of cases. Within a median follow-up period of 72 months, only six patients (0.6%) experienced axillary recurrence and no cases of lymphedema occurred. In their retrospective comparative study, the incidence of arm lymphedema was significantly lower in patients who underwent partial lower ALND compared with those who received conventional ALND (0% vs.11.8%: *p* < 0.0001) (Table 4). While the authors did not perform SLN biopsy before partial lower ALND in this study, it has been reported that 11.5% of SLNs are located above the second ICBN [34]. To date, partial lower ALND is not recommended in international guidelines.

### 6.2. Conservative Axillary Regional Excision (CARE)

Another form of conservative axillary surgery that has emerged is conservative axillary regional excision (CARE). The procedure involves removing SLNs and other palpable suspicious nodes [35]. Total mastectomy with CARE was performed in 587 patients, including not only 379 cN0 patients but also 208 cN+ patients. A median of eight ALNs were removed per patient. Of the patients, 289 (49.2%) patients received NAC or adjuvant systemic therapy and 152 (25.9%) received postoperative radiotherapy. Within a median follow-up period of 5.1 years, axillary recurrence only developed in three (0.5%) patients, while arm lymphedema occurred in twenty (3.4%) patients (Table 4). These findings support the use of the CARE procedure in selected cN+ patients. However, it is worth noting that the CARE procedure is largely identical to SLN biopsy. The American Society of Clinical Oncology (ASCO) recommends that suspicious palpable nodes, irrespective of their dye or radioisotope uptake, be removed as a part of SLN biopsy [36]. Nevertheless, the CARE procedure was performed not only in the cN0 patients but also in the cN+ patients. Previously, we have found that SLN biopsy can be used to assess the axillary nodal status of both cN+ and cN0 breast cancer patients [37].

### 6.3. Conservative ALND with Axillary Reverse Mapping (ARM)

ARM was developed as a surgical procedure to minimize the risk of arm lymphedema by delineating and preserving arm-draining nodes and lymphatics during ALND [38,39]. Since the procedure first emerged, multiple randomized trials have confirmed that ARM is effective in reducing arm lymphedema [40,41,42,43,44,45]. However, there have also been concerns about the oncological safety of ARM due to the not-infrequent involvement of ARM nodes. There are lymphatic interconnections between nodes draining from the upper extremity and nodes draining from the breast [46]. Using the blue dye technique, a study of Boneti et al. [47] observed that lymphatic drainage from the arm and breast rarely converged in SLN and that none of the receiving nodes contained metastases. As a result, the study concluded that preserving the ARM nodes may decrease the incidence of postoperative lymphedema (Table 5).

Although the ARM procedure is generally performed using blue dye, the fluorescent method, which uses indocyanine green (ICG), was developed as an alternative [49]. The fluorescent imaging technique is highly sensitive in identifying ARM nodes. In a study by Noguchi et al. [48], 507 patients with cN0 breast cancer underwent SLN biopsy with the ARM procedure using fluorescence dye. Although ARM nodes were involved in 18 of the 65 patients in which ARM nodes were identified, in 14 (78%) of these patients, the involved ARM nodes were identical to the SLNs identified. Since SLN–ARM nodes should be removed, ARM nodes were ultimately involved in only four (5.7%) patients after SLN biopsy. The authors concluded that, with the exception of positive SLN–ARM nodes, the involvement of ARM nodes in SLN-positive patients is infrequent (Table 5). Nevertheless, indications for ARM in cN0 patients might be limited in light of results from the Z0011 and AMAROS trials [44]. As mentioned above, ALND can be avoided in patients with one to two positive SLN(s) undergoing either BCS with whole breast irradiation [6] or mastectomy with axillary radiation [7]. On the other hand, ARM nodes are involved in a significant proportion of cN+ patients [50]. Therefore, identified ARM nodes with suspected malignancy must be removed even in the ARM procedure. However, it is important to note that ARM lymphatics draining from the upper extremity should be spared as much as possible in order to minimize the risk of arm lymphedema.

A randomized clinical trial was conducted to compare conventional ALND and ARM-guided ALND [45]. To identify the ARM nodes and lymphatics, methylene blue and radioisotope were injected into the upper extremity. A total of 265 patients including pN1-3 patients were randomized into two groups: 127 patients underwent conventional ALND and 138 patients underwent ARM-guided ALND with the aim of preserving ARM nodes and lymphatics. Nevertheless, in the ARM-guided ALND group, after examining identified ARM nodes using fine needle aspiration cytology, ARM nodes were determined to be involved and were subsequently removed in 11 patients (8.5%). In the median follow-up period of 20 months, arm lymphedema had developed in forty-two (33.1%) of the patients who underwent conventional ALND and in seven (5.9%) of the patients who underwent ARM-guided ALND (*p* < 0.001). No patients in either group developed axillary recurrence. The study concluded that while ARM-guided ALND can reduce the incidence of lymphedema, this approach may not be suitable for advanced breast cancer (pN2-3) patients (Table 6).

The iDEntification and Preservation of ARm lymphaTic system (DEPART) is another form of conservative axillary surgery that has been developed [40]. In the study that introduced this procedure, fluorescent dye (ICG) and methylene blue were injected into the upper extremity to identify ARM nodes and the subsequent-echelon nodes. All identified ARM nodes and lymphatics were preserved except SLNs and palpable suspicious nodes. Palpable suspicious ARM nodes (fluorescent or blue nodes) were histologically examined by partial frozen sections and removed if determined to be positive. In the randomized clinical trial, 874 cN+ patients and 480 SLN-positive patients were randomized to either undergo conventional ALND or the DEPART procedure. Postoperatively, both groups of patients received adjuvant chemotherapy, and high-risk patients underwent axillary radiotherapy. Within a median follow-up period of 37 months, arm lymphedema had developed in 18 (3.3%) patients who underwent the DEPART procedure and in 99 (15.3%) of patients who underwent conventional ALND (*p* < 0.001); however, the incidence of axillary recurrence did not differ significantly between the two groups (1.4% vs. 1.2%) [40] (Table 6). Thus, palpable nodes with suspected malignancy should be removed even in ARM-guided ALND.

### 6.4. Tailored Axillary Surgery (TAS) for cN+ Patients

Another surgery that has been developed to decrease the incidence of arm lymphedema is TAS. This approach consists in removing all palpable suspicious lymph nodes together with the blue and radioactive SLNs, ideally performed with image-guided localization of the clipped node to achieve optimal results [51,52,53]. This procedure is performed on cN+ patients, either after NAC or in the upfront surgical setting. Ultimately, TAS aims to turn cN+ patients into cN0 patients primarily through the selective removal of palpable suspicious nodes. Following TAS, axillary radiation is administered to treat any remaining nodal disease. Nodal radiotherapy is effective in achieving local control in patients with low-volume remaining nodal disease.

A randomized clinical trial known as the TAXIS trial is currently underway to investigate the oncological safety and improvement to quality-of-life associated with TAS and axillary radiotherapy in comparison with the use of conventional ALND in patients with cN+ breast cancer. The primary endpoint of this non-inferiority trial is disease-free survival, and secondary endpoints include morbidity and quality of life. Accrual completion is projected for 2025 and the primary endpoint analysis is expected to finish in 2029 [53]. To date, the feasibility of TAS has been confirmed in a pre-planned sub-study involving 296 patients [52]. Of these patients, 125 (42%) received NAC, and 71 (56.8%) of this cohort achieved nodal pCR. In this trial, TAS selectively removed positive lymph nodes and remained much less radical than ALND. In fact, for ALND performed after TAS, additional positive nodes were removed in 70% of patients [52]. Although all of these patients received axillary radiation, the rate of incidence for additional positive nodes was significantly higher compared with corresponding rates from the Z0011 and AMAROS trials [6,7].

## 7. Conclusions

While advancements in axillary surgery for patients with breast cancer have achieved much to reduce the risk of complications such as lymphedema, there are subsets of patients in whom ALND cannot be eliminated. Recently, various forms of conservative axillary surgery have been developed in order to minimize arm lymphedema without increasing axillary recurrence. In the era of effective multimodality therapy, conventional ALND may not be necessary in either cN0 or cN+ patients. Nevertheless, palpable nodes with suspected malignancy must be removed even in conservative axillary surgery because regional radiotherapy is effective in patients with low-volume remaining nodal disease. Further studies with a longer follow-up period are needed to determine the safety of conservative axillary surgery.

## Figures and Tables

**Figure 1 cancers-15-05353-f001:**
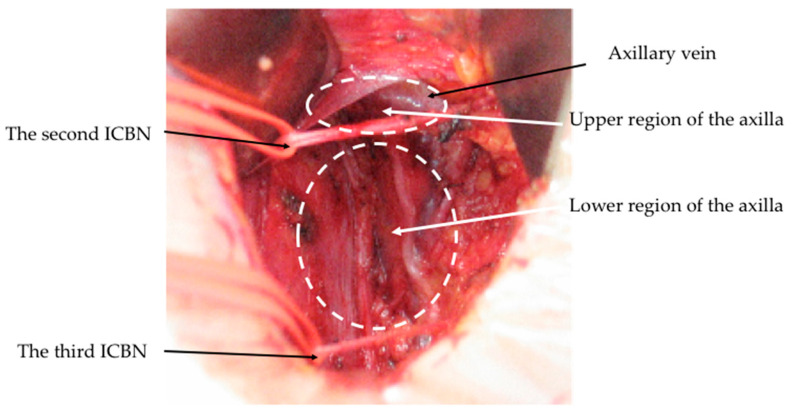
Axillary anatomy after conventional ALND. Lateral thoracic artery and vein were removed; intercostobrachial nerves (ICBNs) were preserved.

**Table 1 cancers-15-05353-t001:** NSABP B-04 trial.

Procedures	(a) cN0 Patients		(b) cN+ Patients	
	Axillary Recurrence	10-Year Survival	Axillary Recurrence	10-Year Survival
ALND	1% (5/362)	58%	1% (3/292)	38%
Regional radiation	3% (11/352)	59%	12% (35/294)	39%
None	18% (65/365)	54%	/	/

ALND: axillary lymph node dissection; cN0: clinically node-negative; cN+: clinically node-positive.

**Table 2 cancers-15-05353-t002:** SLN biopsy alone vs. conventional ALND in cN0, SLN-negative patients.

Trials/Methods	No. of Patients	Radiotherapy	Systemic Therapy	Axillary Recurrence	Survival	Arm Lymphedema
(a) NSABP B-32 trial					8-year ♦	
SLN biopsy alone	2011	82%	84%	0.4%	90.3%	7.5%
SLN biopsy followed by ALND	1975	82%	85%	0.1%	91.8%	14.3% *
(b) Milan trial					10-year ♣	
SLN biopsy alone $	259	100%	47%	0.0%	89.9%	1.0%
SLN biopsy followed by ALND	257	100%	47%	0.8%	88.8%	37% #

$: included 92 patients with positive SLNs who underwent ALND; ♦: overall survival; ♣: cancer-free survival; *: *p* < 0.001; #: *p* < 0.01.

**Table 3 cancers-15-05353-t003:** SLN biopsy alone vs. conventional ALND in cN0, SLN-positive patients.

Trials/Methods	No. of Patients	Radiotherapy	Systemic Adjuvant Therapy	Axillary Recurrence	Arm Swelling
(a) ACOSOG Z0011 trial					
SLN biopsy alone	446	100%	97%	0.9%	2%%
SLN biopsy followed by ALND	445	100%	96%	0.5%%	13%% *
(b) AMAROS trial					
SLN biopsy alone	681	87%	90%	1%%	6%%
SLN biopsy followed by ALND	744	85%	90%	0.5%%	13% #

ALND: axillary lymph node dissection; SLN: sentinel lymph node; *: *p* < 0.0001; #: *p* < 0.0009.

**Table 4 cancers-15-05353-t004:** Partial lower ALND and CARE procedure.

Patients/Methods	No. of Patients	Systemic Chemotherapy	Radiotherapy	Follow-Up Period	Axillary Recurrence	Arm Lymphedema
(a) cN0 patients						
Partial lower ALND	1043	100%	Not reported	72 months	0.6%	0.0%
Conventional ALND	1084	100%	Not reported	120 months	0.1% *	11.8% #
(b) cN0 and cN+ patients						
CARE procedure	587	49.3%	25.9%	5.1 years	0.5%	3.4%

ALND: axillary lymph node dissection; CARE: conservative axillary regional excision; *: not significant; #: *p* < 0.0001.

**Table 5 cancers-15-05353-t005:** The surgical procedures with ARM and the outcome in cN0 patients.

Authors/ Surgical Procedures with ARM	Identification Rate of SLNs	Identification Rate of ARM Nodes or Lymphatics	Crossover Rate between SLN and ARM Nodes	Involved Rate of ARM Node	Axillary Recurrence	Objective Lymphedema
(a) Boneti et al. [47]						
SLN biopsy (n = 220)	97.2% (214/220)	40.6% (87/214)	2.8% (6/214)	0% (0/15)	/	
and/or ALND (n = 47)					/	5.4% (2/51)
(b) Noguchi M, et al. [48]						
SLN biopsy alone (n = 437)	98% (499/507)	63% (321/507)	28% (140/499)		0.9% (4/429)	0.7% (3/429)
ALND after SLN biopsy (n = 70)		93% (65/70)	34% (24/70)	5.7% (4/70) #	2.9% (2/70)	21% (15/70) *

ALND: axillary lymph node dissection; ARM: axillary reverse mapping; SLN: sentinel lymph node; #: involved rate of ARM nodes, except for positive SLN-ARM node; *: *p* < 0.01.

**Table 6 cancers-15-05353-t006:** ARM-guided ALND versus conventional ALND.

Authors/Methods	No. of Patients	Adjuvant Chemotherapy	Radiotherapy	Follow-Up Periods	Axillary Recurrence	Arm Swelling
(a) Yue et al. [45]						
ARM-guided ALND	138	None	Not reported	20 months	0%	5.9%
Conventional ALND	127	None	Not reported	20 months	0% *	33.1% #
(b) Yuan et al. [40]						
ARM-guided ALND	543 &	None	56.5%	37 months	1.4%	3.3%
Conventional ALND	648 &	None	60.6%	37 months	1.2% *	15.3% #

ALND: axillary lymph node dissection; ARM: axillary reverse mapping; *: ns; #: *p* < 0.001. &: thirty-two patients with lost follow-up were excluded from the study, 15 in the study group and 17 in the controlled group.

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
