# Peer review of "Conservative Axillary Surgery May Prevent Arm Lymphedema without Increasing Axillary Recurrence in the Surgical Management of Breast Cancer"

_cancers, 2023, doi:10.3390/cancers15225353_

Round 1
Reviewer 1 Report
Comments and Suggestions for Authors
1. Accepted in minor revision
2. (1) For convenience to readers, the Table 1, 2, 3 were difficulty to read, should summarized in two tables as N(-) and N(+), and Table 4, 5, 6 should summarized to 2 tables.
(2) The author didn't mentioned about update data of omit axillary surgery.
(3) For authors "partial lower ALND", the author many describe more detail in comparison to 4 nodes sampling, or axillary sampling.
Author Response
I have answered to several questions and comments from your.
#1: I have combined Table 4 and Table 5. Otherwise, it is difficult to combine the other Tables. I have improved Tables.
#2: The following sentence has been added.
Recently, nevertheless, axillary staging is not indicated for staging, local control, or determining the need for systemic therapy in 70 or more older women with cT1-2, cN0, luminal type breast cancer [14].
#3: The following sentence has been added.
In this procedure, the level I axillary nodes caudal to the nerve was excised, while retracting the second ICBN by using tape. It is different from four nodes sampling where hard or enlarged axillary nodes were selectively removed from the axillary tissue.
Reviewer 2 Report
Comments and Suggestions for Authors
The incidence rate of breast cancer-related lymphedema is decreasing due to the big change in therapeutic approaches for breast cancer. The incidence varies for axillary lymph node dissection and sentinel node procedures, respectively. In recent years there has been a reduction in aggressive treatments such as axillary dissection for the treatment of breast cancer. Breast cancer-related lymphedema is an extremely dreaded complication after breast cancer treatment. This Review article aims to update the reader with the current understanding of these subjective lymphedemas after breast cancer treatment without increasing axillary recurrence in the surgical management of breast cancer
The paper presents points that need to be improved:
The authors state in line 118 “The axilla is anatomically divided into upper and lower parts, as separated by the
second intercostobrachial nerve (ICBN)”. The anatomical classification of the axillary lymph nodes
has undergone over time variations, particularly related
to their clinical implications. There are numerous publications discussing the prevention of lymphedema during axillary surgery. Please review the bibliography by improving it, I recommend reading these publications:
- De Luca A et al Retrospective Evaluation of the Effectiveness of a Synthetic Glue and a Fibrin-Based Sealant for the Prevention of Seroma Following Axillary Dissection in Breast Cancer Patients. Front Oncol. 2020 Jul 17;10:1061. doi: 10.3389/fonc.2020.01061. PMID: 32766138; PMC ID: PMC7379884.
- Cirocchi R et al: New classifications of axillary lymph nodes and their anatomical-clinical correlations in breast surgery. World J Surg Oncol. 2021 Mar 29;19(1):93. doi: 10.1186/s12957-021-02209-2. PMID: 33781279; PMCID: PMC8008673.
- The figures need to be improved in graphics and style.
- Both the objectives, methods of this research are broad
- The discussion part needs to be improved, you should add a report part about surgical treatment and prognosis to make your study interesting for the readers.
To conclude, this is a potentially good review. However, it needs to cropped across to make it more concise and relevant to the research topic. I would advise authors to revisit their literature search at latest evidence published in the past 3 years so that this review can be strengthened. Good luck!
Author Response
#1. The following sentence has been added.
although lymph nodes lateral to the lateral thoracic vein are more specific to the lymphatic drainage of the upper extremity than to the drainage of the breast [31].
#2. The article of Cirocchi et al. [29] has been added in the reference. On the other hand, the prevention of seroma following axillary dissection (De Luca A et al) is another issue.
#3. The figure 1 has been added.
Reviewer 3 Report
Comments and Suggestions for Authors
The article by Noguchi et al. presents an up-to-date review on axillary lymph node management in breast cancer patients.
I recommend performing major revisions.
1. line 27 - "There is a significant portion of breast cancer patients in whom ALND cannot be avoided." I can´t agree with this information, the most of breast cancer patients are eligible for SLNB only, so ALND can be avoided.
2. line 84... - The actual approach to breast cancer patients after NAC is not clearly listed in this chapter, there is no information about targeted axillary dissection or specific problems with sentinel lymph node biopsy after NAC. Also, the actual opinions on residual nodal disease among breast cancer specialists also differ from your article - look to St. Gallen conference.
3. Current indications for ALND are also not accurate - according to NCCN guidelines you can avoid ALND in cN+ patients up to 3 positive lymph nodes.
4. Partial lower ALND is also called level I axillary dissection, I recommend performing another review and compare the different information about this method. Also, it is not recommended in any international guidelines.
5. References - there are a lot of citations of your articles, I recommend adding more bigger international studies.
Author Response
#1. The following sentence has been removed.
There is a significant portion of breast cancer patients in whom ALND cannot be avoided.
#2. The sentence has been changed and added as follows.
The most of breast cancer patients are eligible for SLN biopsy only, so ALND can be avoided. However, there is subsets of patients in whom ALND remains necessary [9]. According to the NCCN Guidelines, ALND of level I and II nodes is recommended to patients with biopsy proven axillary metastases (in those who did not receive NAC) or who have residual disease after NAC [25]. In the St. Gallen International Consensus conference in 2021 [26], nevertheless, there was controversy about lower residual disease after NAC (for instance, a micrometastasis or isolated tumor cells in one of three SLNs). On the other hand, Beck and Morrow [9] have summarized the current indications for ALND as follows.
#3. The following sentence has been added.
Although SLN biopsy has a >10% false-negative rate (FNR) when performed after NAC [18,19], the FNR can be reduced by marking biopsied lymph nodes to document their removal, using dual tracer, and by removing three or more SLNs (targeted axillary dissection) [20].
#4. The sentence has been changed and added as follows.
According to the NCCN Guidelines, ALND of level I and II nodes is recommended to patients with biopsy proven axillary metastases (in those who did not receive NAC) or who have residual disease after NAC [25]. In the St. Gallen International Consensus conference in 2021 [26], nevertheless, there was controversy about lower residual disease after NAC (for instance, a micrometastasis or isolated tumor cells in one of three SLNs). On the other hand, Beck and Morrow [9] have summarized the current indications for ALND as follows.
#5. Partial lower ALND is not same as level I axillary dissection. It removed a lower region of level I axillary tissue. The following sentence has been added.
In this procedure, the level I axillary nodes caudal to the nerve was excised while retracting the second ICBN using tape. It is different from four nodes sampling where hard or enlarged axillary nodes were selectively removed from the axillary tissue.
#6. The following sentences and references have been added.
The other randomized trials also have supported these findings [16,17].
- Galimberti V, Cole BF, Zurrida S, et al. Axillary dissection versus no axillary dissection in patients with sentinel-node micrometastases (IBCSG 23-01): a phase 3 randomised controlled trial. Lancet Oncol2013; 14:297–305. doi: 10.1016/S1470-2045(13)70035-4.
- Sávolt Á, Péley G, Polgár C, et al. Eight-year follow up result of the OTOASOR trial: the optimal treatment of the axilla - surgery or radiotherapy after positive sentinel lymph node biopsy in early-stage breast cancer: a randomized, single centre, phase III, non-inferiority trial. Eur J Surg Oncol2017; 43:672–679. doi: 10.1016/j.ejso.2016.12.011.
- KuehnT,Bauerfeind I, Fehm T, et al. Sentinel-lymph-node biopsy in patients with breast cancer before and after neoadjuvant chemotherapy (SENTINA): a prospective, multicentre cohort study. Lancet Oncol 2013; 14:609–618. doi: 10.1016/S1470-2045(13)70166-9.
- 19. Boughey JC, Suman VJ, Mittendorf EA, et al. Sentinel lymph node surgery after neoadjuvant chemotherapy in patients with node-positive breast cancer: the ACOSOG Z1071 (Alliance) clinical trial. JAMA2013;310:1455–1461. doi: 10.1001/jama.2013.278932.
- CaudleAS, YangWT, Krishnamurthy S, Mittendorf EA, Black DM, Gilcrease MZ, Bedrosian I , Hobbs BP, DeSnyder SM, Hwang RF, Adrada BE, Shaitelman SF, Chavez-MacGregor M, Smith BD, Candelaria RP, Babiera GV, Dogan BE, Santiago L, Hunt KK, Kuerer HM. Improved Axillary Evaluation Following Neoadjuvant Therapy for Patients With Node-Positive Breast Cancer Using Selective Evaluation of Clipped Nodes: Implementation of Targeted Axillary Dissection. J Clin Oncol. 2016; 34:1072-1078. doi: 10.1200/JCO.2015.64.0094.
26.Burstein HJ, Curigliano G, Thürlimann B, Weber WP, Poortmans P, Regan MM, Senn HJ, Winer EP, Gnant M; Panelists of the St Gallen Consensus Conference. Customizing local and systemic therapies for women with early breast cancer: the St. Gallen International Consensus Guidelines for treatment of early breast cancer 2021. Ann Oncol. 2021; 32:1216-1235. doi: 10.1016/j.annonc.
Reviewer 4 Report
Comments and Suggestions for Authors
I think this review paper is concise and easy to read. There is one point that I would like to add, which is related to Section 4, and I think you should add the following review.
For breast cancer cases with clinically positive lymph nodes at the pre-treatment time point, if the lymph nodes become negative after NAC, SLN biopsy may be indicated, please review the results of these SLN biopsy clinical trials (identification rate, false negative rate, etc.).
Author Response
The following sentences have been added:
In the SENTINA study [18], the detection rate of SLNs was 80.1% and the overall FNA was 14.2%, and in ACOSOG-Z1071 trial [19], the detection rate of SLNs was 92.9% and the FNA was 12.6%. However, TAD followed by ALND was performed with an FNR of 2.0% [20].
Round 2
Reviewer 3 Report
Comments and Suggestions for Authors
Thank you for the opportunity to do a review of the revised version. Unfortunately, there are still some inaccuracies.
1. "According to the NCCN Guidelines, ALND of level I and II nodes is recommended to patients with biopsy proven axillary metastases (in those who did not receive NAC) or who have residual disease after NAC [25]."
In patients with macrometastasis in SLN (without NAC), there are conditions from study Z0011 to avoid ALND - look again to NCCN guidelines.
2. "Although SLN biopsy has a >10% false-negative rate (FNR) when performed after NAC [18,19], the FNR can be reduced by marking biopsied lymph nodes to document their removal, using dual tracer, and by removing three or more SLNs (targeted axillary dissection) "
Sorry, but it is not clear, what really is targeted axillary dissection.
3. "#5. Partial lower ALND is not same as level I axillary dissection. It removed a lower region of level I axillary tissue. The following sentence has been added."
Ok, but it is necessary to cite guidelines or any metaanalysis recommending this procedure because in my opinion it is not recommended in any international guidelines.
4. Finally, there are still not sufficent information about targeted axillary dissection and targeted lymph node dissection in the article, while these are recommended procedure in patients with cN1 before NAC and ycN0 after NAC.
Round 3
Reviewer 3 Report
Comments and Suggestions for Authors
The revised version is better, but still, a few inaccuracies are present.
Ad 1 - Study Z0011 is a part of the diagram of NCCN guidelines, so the sentence should be edited.
Ad 2 and 4 - The lymph node could be localised by many other localisation methods, not only an iodine seed - e.g. magnetic seed, carbon suspension, Savi scout etc - please add this information according to the recent trials, because this is an area of recent investigation by many authors.
Ad 3 - ok.
Also, the authors did not add my recommendation about TLND (targeted lymph node dissection) in the article, so I think this should be added along with a better description of TAD as I mentioned above.
